# Compressive Properties of Functionally Graded Bionic Bamboo Lattice Structures Fabricated by FDM

**DOI:** 10.3390/ma14164410

**Published:** 2021-08-06

**Authors:** Zhou Wen, Ming Li

**Affiliations:** 1School of Mechanical Engineering, Xi’an University of Science and Technology, Xi’an 710054, China; wenz@dgpt.edu.cn; 2Department of Media Communication, Dongguan Polytechnic, Dongguan 523808, China

**Keywords:** functionally graded material, fused deposition modeling, lattice structure, mechanical property, bionic design

## Abstract

Bionic design is considered a promising approach to improve the performance of lattice structures. In this work, bamboo-inspired cubic and honeycomb lattice structures with graded strut diameters were designed and manufactured by 3D printing. Uniform lattice structures were also designed and fabricated for comparison. Quasi-static compression tests were conducted on lattice structures, and the effects of the unit cell and structure on the mechanical properties, energy absorption and deformation mode were investigated. Results indicated that the new bionic bamboo structure showed similar mechanical properties and energy absorption capacity to the honeycomb structure but performed better than the cubic structure. Compared with the uniform lattice structures, the functionally graded lattice structures showed better performance in terms of initial peak strength, compressive modulus and energy absorption.

## 1. Introduction

Functionally graded materials (FGMs) have received extensive attention for application in various fields including aerospace [1,2,3], biomechanics [4,5] and energy absorption [6,7,8,9]. They can be characterized by their mechanical properties as the composition and structure gradually change over volume. For specific function and application, several traditional manufacturing methods have been employed to fabricate FGMs; these methods include plasma spraying, physical vapor deposition and powder metallurgy [10]. In contrast to traditional manufacturing methods, additive manufacturing (AM) has significant advantages in fabricating complex lattice structures, as it can overcome the limitations of traditional manufacturing methods by adding materials from a bottom-up, layer-by-layer approach [11].

For FGMs, most researchers focus on manufacturing multi-material compositions or designing structures. Fused deposition modeling (FDM), selective laser melting (SLM) and electron beam melting (EBM) are the main multi-material AM techniques that have been extensively applied to fabricate functionally graded composites [1]. Khalil et al. fabricated four different material depositions through a multi-nozzle heterogeneous system by FDM [12]. Attar et al. studied the mechanical properties of FGMs by comparing the compression of pure titanium (CP–Ti) and porous Ti–TiB composite parts fabricated by SLM, and they found that the yield strength and elastic modulus of porous CP–Ti parts are much lower than those of porous Ti–TiB samples [13]. Yang et al. fabricated Ti–Mo gradient material by EBM, and their material demonstrated high-temperature resistance [14]. Various metals and plastics have been selected to manufacture FGMs, such as Ti, polylactic acid (PLA) and acrylonitrile–butadiene–styrene (ABS), but the use of polyether ether ketone (PEEK) has received little attention. PEEK is a colorless organic thermoplastic polymer with good mechanical properties; it is used in the automotive, aerospace, and medical implant industries due to its good biocompatibility and elastic modulus, which is close to that of human bones [15,16]. Compared with other filament materials, such as PLA and ABS, PEEK has better mechanical properties, and its elastic limit and tensile strength were experimentally investigated, reaching 50.8 and 56.6 MPa, 122% and 108% higher than the values for ABS [17]. However, articles on FGMs made of PEEK are limited. Therefore, the mechanical properties of FGMs made of PEEK should be studied.

Thin-walled structures, strut-based structures and surface-based structures in which the geometry of lattice structures can be designed and modeled for special functions have attracted considerable attention amongst researchers [18]. Wang et al. improved the mechanical properties of functionally graded thin-walled structures with a honeycomb unit by topology optimization of density distribution [19]. Li et al. conducted experiments and finite element analysis to demonstrate that desired mechanical properties of FGMs can be achieved by designing and generating variable-density gyroid (surface-based) structures [20]. Rinoj et al. investigated the mechanical properties of functionally graded and uniform density Kagome (strut-based) structures made of ABS by FDM, and they found that the functionally graded Kagome structure provides 35% more energy absorption than the uniform density structure [21].

At present, many FGMs with varying traditional geometrical structures, such as square tubes, honeycomb cells and cuboctahedron cells, have been investigated [22]. However, the design of novel structures for FGMs has been largely ignored. Some researchers designed thin-walled structures inspired by biological structures, as they have excellent properties and ingenious frames to adapt to extreme conditions after billions of years of evolution [23]. Chen et al. designed three new types of bionic tubes based on the structure of bamboo and manufactured them using stainless steel by lathe machining, and they experimentally studied the deformation modes and energy absorption capacity of the new structures [24]. Yin et al. designed new bionic thin-walled structures inspired by horsetails and investigated their mechanical properties and energy absorption capacity by nonlinear finite element analysis [23]. Therefore, the natural structures of FGMs applied to energy absorption should be analyzed.

In this study, a new structure inspired by bamboo was designed, and two other geometry structures were imported for comparative analysis. These FGM structures were designed with continuously and linearly changing diameter of lattice along the build direction. Uniform and FGM structures were fabricated with PEEK by FDM, and quasi-static compressive tests were carried out to study the mechanical properties and energy absorption.

## 2. Materials and Methods

### 2.1. Design of Bionic Structures Based on Bamboo

Bamboo, like wood, is a natural composite material with a high strength-to-weight ratio, good elasticity and stability [25]. Moreover, it is considered the most effective structure in nature, and its strength is three times higher than that of steel [26]. As shown in Figure 1a, bamboo consists of joints and stems, and vascular bundles are scattered throughout its stems. In terms of the macrostructure of bamboo, the thickness of the tube decreases significantly along the axial direction from the top to the bottom (Figure 1b). Two types of bionic structures inspired by the geometry of bamboo were designed to improve the mechanical properties and energy absorption capacity of the FGM structure. One was designed with a uniform strut diameter, and the other (Figure 1c) was designed to vary with strut diameter continuously and linearly.

### 2.2. Design and Production of Lattice Structures

SolidWorks 2014 software (Dassault Systèmes SolidWorks Co., Waltham, MA, USA) was used to design uniform and FGM lattice structures. Three different unit cells, named cubic unit cell, honeycomb unit cell and bamboo unit cell, were imported to design the uniform lattice structures and FGM lattice structures (Figure 2). The size of the unit cell in the cubic lattice was 9.53 mm× 9.53 mm × 9.53 mm, whereas the circumscribed circle diameter of the unit cell in the honeycomb lattice was 12 and 9.53 mm in height. Similar to the honeycomb unit cell, the height of the unit cell in the bamboo lattice was 9.53 mm, and its diameter was 10.39 mm. In all unit cells, the cross section of solid struts was circular (circular portion), with a radius of 1.5 mm. As shown in Figure 2, the number of unit cells in the cubic lattice structure in three directions (X, Y and Z) was 6 × 6 × 6. For the honeycomb and bamboo lattice structures, the unit cells were distributed in an alternate permutation in the X–Y plane, with a total of 31 in the plane. The unit cell number was consistent with the cubic lattice structure in the Z direction. In the uniform lattice structures, the cell strut diameter was 3 mm, and the nodal joints had no sharp ends. In contrast to the uniform lattice structures, the cell strut diameters of the FGM lattice structures were designed to vary from the bottom to top along the Z direction. The cell strut diameters changed linearly and continuously in the building direction (Z direction) and were 2–2.33, 2.33–2.67, 2.67–3, 3–3.33, 3.33–3.67 and 3.67–4 mm for the six structural layers.

The uniform lattice structures and FGM lattice structures were designed using SolidWorks 2014, and PreForm software (Formlabs Inc., Somerville, MA, USA) was employed to slice and prepare them for 3D printing. To improve precision, the layer thickness of slices was set at 0.1 mm. The filling ratio of 100% was selected to obtain improved mechanical performance. Specimens for each design were manufactured with PEEK(PEEK880G), which was supplied by HongKai Corporation (HongKai Co., Leqing, China), by using an OMNISY H600 FDM machine form (Feifanshi Co., Xi’an, China). This machine was different from the traditional FDM machine for PLA or ABS production. The OMNISY H600 FDM machine consisted of a heated bed to prevent warpage at the edges of PEEK printed parts. The manufacturing process parameters are shown in Table 1.

### 2.3. Mechanical Testing

A universal mechanical testing machine (LE 3504, LiShi Co., Shanghai, China) with 50 KN load capacity was used for mechanical compressive tests. Before the tests, uniform and FGM structure specimens were placed on the center of the bottom plate. The bottom plate was continuously pressed, and the top plate kept moving towards the bottom in compression (Figure 3). Following the ASTM standard D695–15, the crosshead loading rate was set at 0.1 mm/s, and displacement was measured by the crosshead movement. Deformation images of specimens were recorded or captured by using a camera.

### 2.4. Finite Element Modeling and Analysis

The deformation behavior of different lattice structures can be simulated based on the finite element method (FEM). In this study, the finite element model was established by using ABAQUS software with 536,382 nodes of a 10-node quadratic tetrahedron (C3D10). The top and bottom plates were established as analytically rigid in the finite model, the specimen was in close contact with the bottom plate and the six freedoms of the top plate were completely fixed. At the same time, the bottom plate moved freely in the Y-direction at a speed of 0.1 mm/s and fixed in other directions. There were no boundary constraints applied to the sides of the lattice models. The stress–strain curve of the same batch of specimens obtained by the tensile test is shown in Figure 4. The material mechanical parameters used in the finite element model are as follow: Young’s modulus *E* = 1.4 GPa, Poisson’s ration *v* = 0.4, density ρ = 1.3 g/cm^3^ and yield strength *σ**_s_* = 24 MPa.

## 3. Results and Discussion

### 3.1. Mechanical Properties and Energy Absorption

Quasi-static compressive tests of FGM and uniform structure specimens were conducted along the specimens’ building direction (Z direction). Force and displacement data of the tests were recorded by a computer and converted to stress and strain. The stress and strain data of tests were imported to Origin 9.1 software. Figure 5 shows the stress–strain curves of the tests for different lattice structure specimens. To evaluate these structures’ mechanical properties and the energy absorption ability of different lattice structures, we obtained these quantities with the following equations [27]:$$\bar{\rho }=\frac{\rho }{\rho_{s}}$$
$$E=\frac{\delta_{y}}{0.02}$$
$$W_{c}=\int_{\varepsilon =0}^{\varepsilon =0.4}\delta \left(\varepsilon \right)d_{\varepsilon }$$
where $\bar{\rho }$ is the relative density of the lattice structure, ρ is the density of the lattice structure in g/cm^3^, $\rho_{s}$ is the density of PEEK (1.29 g/cm^3^), E is the compressive modulus in MPa,$\;\delta_{y}$ is the yield stress in MPa obtained at 0.02 of strain (linear part of the stress–strain curve), and $W_{c}$ is the energy absorption per unit volume calculated by numerically integrating the stress–strain curves up to the strain of 0.4 due to the absence of stabilized densification (Figure 5) [27]. The energy absorption per unit volume of different lattice structures offers useful insight for application of these structures in impact and energy absorption applications. The compression properties for the six lattice structure specimens are summarized in Table 2. Only one stress peak was observed in all the uniform lattice structures, and the strain was less than 0.06. For the uniform lattice structures, their initial peak strength was about two times that of FGM with the same unit cell (Table 2). The uniform lattice structures demonstrated brittle behavior with high initial peak strength and small strain in accordance with the image from the digital camera. When the uniform structure specimens were compressed to a strain of 0.05, they were squeezed out of the plate with high pressure in the Z direction, and the test was ended. Compared with the uniform lattice structures, all of the FGM lattice structures obtained multiple peaks in their stress–strain curve. The stress–strain curves started with the smallest peaks corresponding to the thinnest strut diameter in the first layer. The subsequent peaks were increasingly higher, caused by the stronger resistance to the compression load with thicker strut diameter. The last peak was always the maximum peak before densification, and it was three to four times the initial peak. As evidenced by the presence of more peaks in their stress–strain curves, the FGM lattice structures demonstrated better energy absorption capacity than the uniform lattice structures (Table 2). The FGM lattice structures exhibited better energy absorption capacity by more than threefold than the uniform lattice structures with the same unit cell under similar relative density. The compressive modulus from the FGM lattice structures was higher than those from the uniform lattice structures, but the gap between them was narrower than those in terms of initial peak strength or energy absorption.

The unit cell is another critical factor influencing the mechanical properties of lattice structures. As shown in Table 2, the unit cell of bamboo had the highest compression modulus in both the FGM lattice structures and uniform lattice structures amongst the three different unit cells, and tTABhe value of the honeycomb unit cell was close to that of bamboo. The honeycomb unit cell showed slight advantages in initial peak strength and enLEergy absorption, and those values of the honeycomb FGM lattice structures were less than 10% higher than those of the bamboo FGM lattice structures. The cubic unit cell demonstrated the lowest initial peak strength, compression modulus and energy absorption, which indicated that it had the worst mechanical properties and energy absorption ability amongst these unit cells.

### 3.2. Deformation Modes

Figure 6 shows the deformation patterns of the specimens with different lattice structures at varying strain levels under compressive loading. In the primary stage of compression, elastic bending was the main deformation mode of the struts for all structures. Parts in the bottom or top of struts suffered the maximum shear stress perpendicular to the manufacturing direction, and fractures almost occurred in the bottom or top of the vertical struts near joints (Figure 7a). It is also clear from the FE simulation (Figure 8) that the struts near joints suffered the maximum stress and were most likely to fail. In addition, the fracture surface was regularly perpendicular to the manufacturing direction as shown in Figure 7b, as the peer raster overcoming the low bonding strength from neighbor layers was easier than to collapse the raster along the building direction.

The strain of the uniform lattice structures was approximately 0.05 (Figure 6), which was considerably lower than that of the FGM lattice structures. The uniform lattice structures exhibited brittle behavior with low strain, and this result was consistent with the findings of morphological analysis that the uniform lattice structures presented brittle characters as the fracture surface of struts was smooth with shallow dimples (Figure 7c). For the FGM lattice structures, the fracture surface of struts was coarser with deeper dimples than the uniform lattice structures (Figure 7d), and they exhibited ductile behavior. As shown in Figure 6, for the FGM lattice structures, most initial collapses occurred in the first layer, namely the area of red solid dots. As compression continued, the structures crushed sequentially from the top layer to the bottom layer. The deformation behavior of the FGM lattice structures for most samples was consistent with the FE simulation in Figure 9. Given that the strut diameter became thicker from top to bottom, the bottom layer had more resistance ability to pressure than the upper layers. However, an exception was sample 2 of the cubic FGM structure (Figure 6b). The initial collapse layer of this sample was different from those of other FGM lattice structure samples, which occurred in the first and second layers almost simultaneously. By comparing the stress–strain curve of this sample with that of other cubic FGM structure samples in Figure 6b, the results showed the absence of a stress peak between the first peak and the third peak. Moreover, the first stress peak was higher than that of other samples with the same unit cell. In particular, sample 2 of the cubic FGM structure presented a stronger initial peak strength than the other samples, and it lost the ability to absorb energy and resistance of pressure in the second stage of compression. Hence, the stress–strain curve was consistent with the deformation images in Figure 6b. The different deformation modes of sample 2 of the cubic FGM were mostly caused by the defects of struts in the second layer, and these defects reduced the second layer’s resistance ability to the level of the first layer. For the uniform lattice structures with the same unit cell, their strut diameter was almost the same. Therefore, each layer of the uniform lattice structures performed similarly to resist pressure. In Figure 6a,c,e, the initial collapse layer occurred randomly in the uniform lattice structures, such as the first layer, the second layer and the fifth layer. This was caused by manufacturing defects that occurred randomly, and the uniform lattice structures were initially deformed and crushed in the weakest zone. Stringing was observed in Figure 7e, and it was due to the residual molten plastic from the extrusion head that continued to flow as it moved after the end of one layer’s printing. The rough edge and hole, as shown in Figure 7f, were the other main defects that occurred in the manufacturing process. These manufacturing imperfections detected in struts could generate stress concentration and cause strut failure. Once one strut fails, the stress is redistributed to other struts in the same layer and other struts suffer more stress, leading to the collapse of the entire layer. Wu et al. reported that changing the printing speed or building orientation can reduce the stringing. Wang et al. found that mechanical properties and surface quality can be significantly affected by printing temperature and layer thickness. The mechanism of manufacturing defect formation is very complex, and further research will be carried out in the future.

## 4. Conclusions

This study investigated the mechanical properties and energy absorption capacity of uniform and FGM lattice structures of cubic, honeycomb and bamboo cells. Firstly, bamboo uniform and FGM lattice structures were bionic designs inspired by bamboo, and the two other uniform and FGM lattice structures were designed based on the geometry of a cube and honeycomb. Secondly, these specimens were manufactured by FDM technology. Quasi-static compressive tests were carried out on these specimens, and their test values and stress–strain curves were obtained. Their mechanical properties and energy absorption capacity were calculated by the values recorded from the tests. Finally, the fracture surfaces were observed, and deformation modes were analyzed. The following conclusions were drawn from this study.

The unit cell plays an important role in mechanical properties. Bamboo FGM exhibited the highest compressive strength amongst the materials, whereas cubic uniform demonstrated the lowest compressive strength. For initial peak strength, the bamboo uniform and honeycomb uniform structures performed much better than the other unit cells.

Compared with the FGM lattice structures and uniform lattice structures, the uniform lattice structures showed limited energy absorption capacity due to their limited strain in the compression tests. The bamboo FGM and honeycomb FGM structures exhibited similar energy absorption capacity, which was much higher than that of the cubic FGM structure.

For the FGM lattice structures, layer-by-layer crushing was the major failure model beginning with the thinner strut layer followed by the thicker strut layer in sequence. By contrast, the initial collapse of the uniform lattice structures randomly occurred in the layers, and this phenomenon was mainly caused by the struts’ defects generated in the manufacturing process.

SEM results demonstrated that some imperfections were generated in production, such as stringing, hole and rough edge. These defects could cause stress concentration and develop into strut failure and crushing of the entire layer.

## Figures and Tables

**Figure 1 materials-14-04410-f001:**
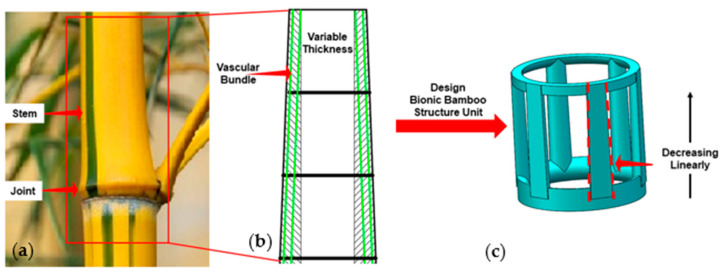
Bionic design inspired by bamboo. (**a**) bamboo in nature, (**b**) longitudinal section, (**c**) bionic bamboo lattice unit.

**Figure 2 materials-14-04410-f002:**
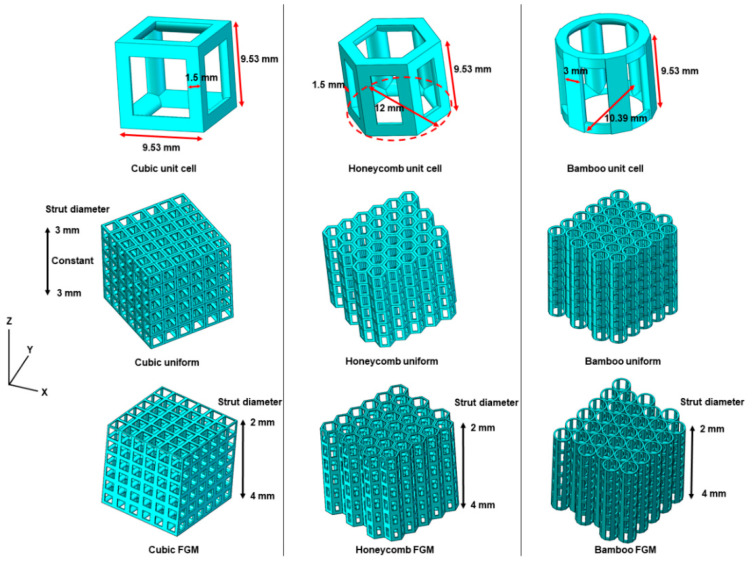
Design of uniform and FGM lattice structures in the building direction.

**Figure 3 materials-14-04410-f003:**
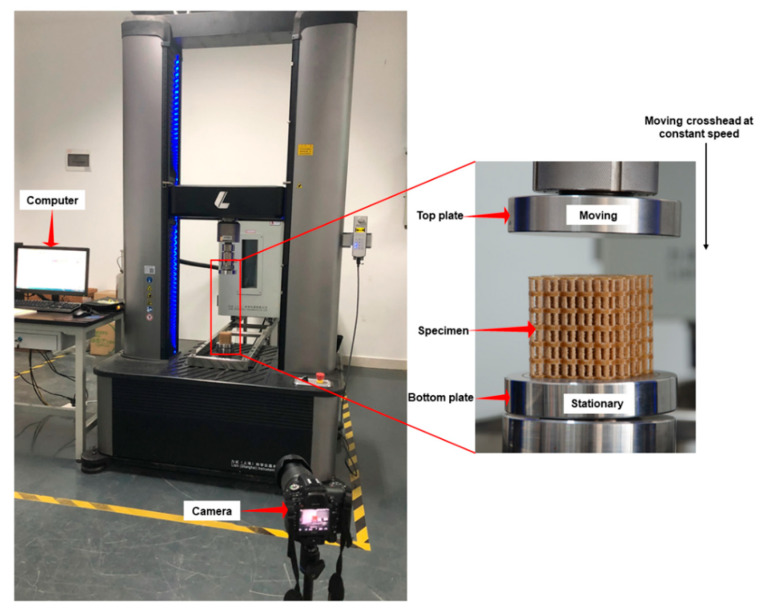
Experimental setup used for compression tests.

**Figure 4 materials-14-04410-f004:**
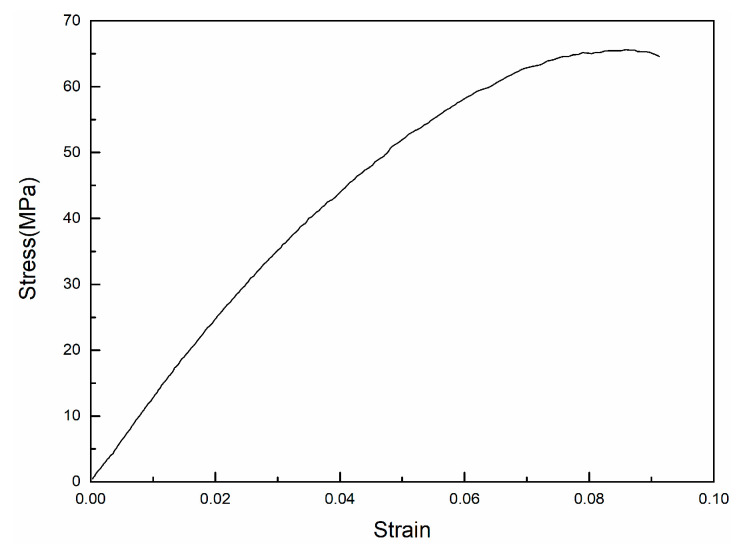
Tensile stress–strain curves of printed PEEK.

**Figure 5 materials-14-04410-f005:**
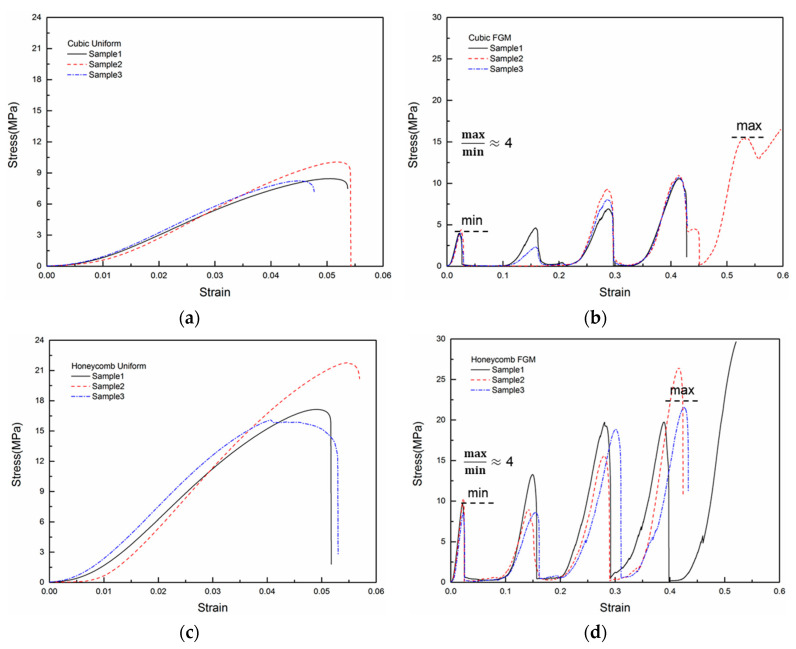
Compressive stress–strain curves of lattice structures: (**a**) cubic uniform (**b**) cubic FGM (**c**) honeycomb uniform (**d**) honeycomb FGM (**e**) bamboo uniform and (**f**) bamboo FGM.

**Figure 6 materials-14-04410-f006:**
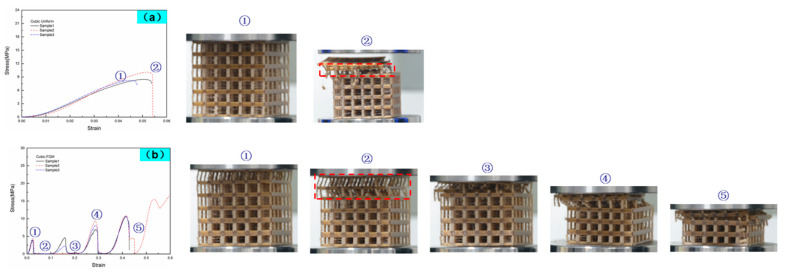
Deformation process of lattice structures: (**a**) cubic uniform, (**b**) cubic FGM, (**c**) honeycomb uniform, (**d**) honeycomb FGM, (**e**) bamboo uniform and (**f**) bamboo FGM.

**Figure 7 materials-14-04410-f007:**
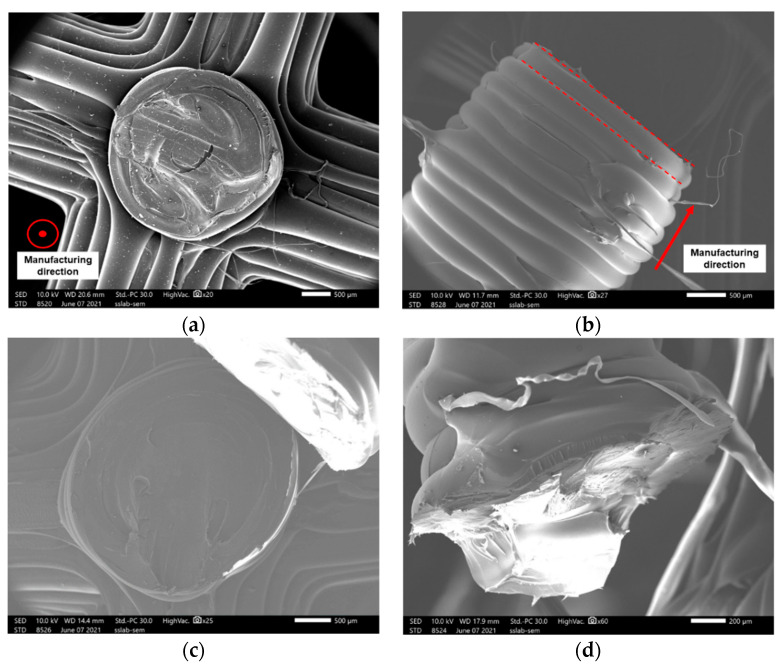
Fracture surface of specimens observed under SEM after compression tests: (**a**) bamboo FGM strut, (**b**) bamboo uniform strut, (**c**) cubic FGM strut, (**d**) honeycomb FGM strut, (**e**) honeycomb uniform strut and (**f**) cubic FGM strut.

**Figure 8 materials-14-04410-f008:**
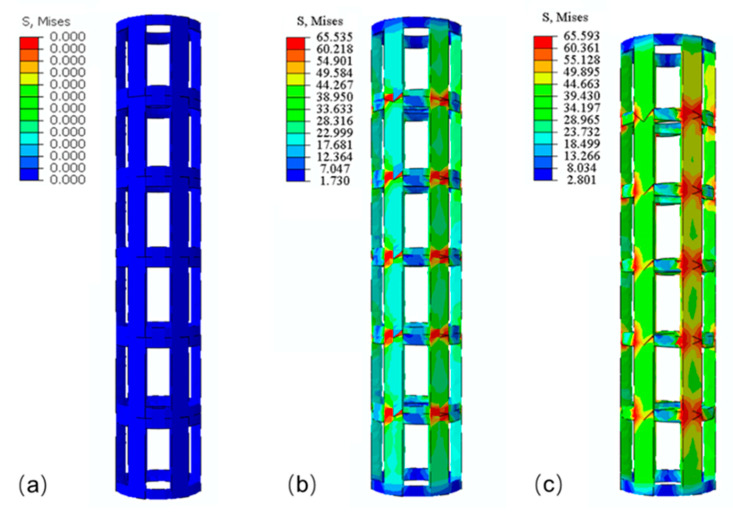
Evolution of deformation behavior of the bamboo uniform lattice structures predicted by FEA: (**a**) ε=0, (**b**) ε=0.01, (**c**) ε=0.03.

**Figure 9 materials-14-04410-f009:**
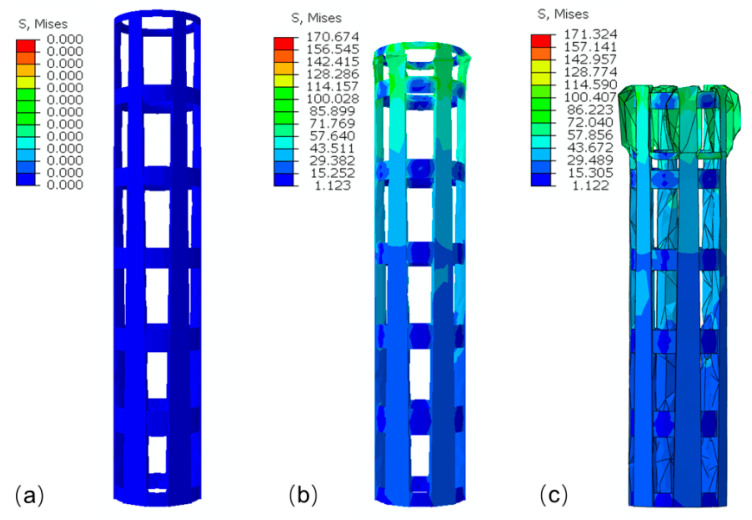
Evolution of deformation behavior of the bamboo FGM lattice structures predicted by FEA: (**a**) ε=0, (**b**) ε=0.07, (**c**) ε=0.15.

**Table 1 materials-14-04410-t001:** Manufacturing parameters of the FDM machine.

FDM Parameters	Values
Printing speed (mm/s)	40
Printing temperature (°C)	400
Bed temperature (°C)	120
Cooling fan (%)	30

**Table 2 materials-14-04410-t002:** Compressive properties of the PEEK specimens with six types of lattice structures.

Lattice Structure	Cubic Uniform	Cubic FGM	Honeycomb Uniform	Honeycomb FGM	Bamboo Uniform	Bamboo FGM
Initial peak strength (MPa)	8.5 ± 1	4.5 ± 0.3	19 ± 3	9 ± 1	18 ± 3	9 ± 1
Compressive modulus (GPa)	0.15 ± 0.01	0.18 ± 0.01	0.31 ± 0.05	0.44 ± 0.03	0.32 ± 0.04	0.45 ± 0.01
Energy absorption (J/cm^3^) (up to strain of 0.4)	0.22 ± 0.03	0.68 ± 0.02	0.51 ± 0.05	1.86 ± 0.09	0.28 ± 0.06	1.72 ± 0.06
Relative density	0.189	0.193	0.233	0.237	0.206	0.211

## Data Availability

Data sharing is not applicable.

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
