# Peer review of "Compressive Properties of Functionally Graded Bionic Bamboo Lattice Structures Fabricated by FDM"

_materials, 2021, doi:10.3390/ma14164410_

Round 1

Reviewer 1 Report

The presented paper provides an insight into quasi-static compressive behaviour of additively manufactured lattices based on uniform or graded unit cells inspired by natural bamboo and commparison to some other unit-cell geometries. Although the field of bionic design for structural applications is worth investigation, the impact of the presented paper in the current form is low.

Overall, the paper is rather causal without the required level of scientific depth and context in this field of mechanical science. In my opinion, the experimental procedures and data treatment are seriously flawed rendering the paper unpublishable in the current form and I recommend its rejection.

My specific comments are:

L85 - you exagerate the properties of bamboo in a manner inappropriate for a scientific paper but common in popular sources; the formulation is, furthermore, questionable because absolute magnitude of bamboo strength (which one - tensile, shear, compressive, etc.) might not be three times higher - did you mean some specific strength?

I found no comments regarding the size effect in the studied lattices and a clarification towards RVE and its implications on the formulated conclusions.

No mechanical properties of the base material used for production of the lattices were given, particularly the mechanical properties resulting from 3D printing of such slender beams at this scale level. When reviewing your results, I suggest to concentrate on deformation modes causing the failure of your lattices.

Common design of loading device was used in the experiments. Given the crushing response of your samples and according to my experience with similar tests of lattices based on 3D printed PMMA-like material, you have to comment on the typical force magnitudes during the experiments and provide a representative plot in relation to sensitivity of the used 50 kN load cell.

It is common in this field that displacements and strains are measured optically to treat the problems arising from the nature of such porous beam-based constructs influencing the boundary conditions. This would also help to cope with the errors introduced by the deformation of the loading device itself which inevitably influences the acquired results, particularly if you want to estimate the elastic characteristics.

I do not understand the term "Awag" in Figure 3.

Given the deformation and failure modes of your samples, it is necessary to provide comments on the reason for estimation of the deformation energy absorption.

The engineering method for estimation of elastic modulus in compression is unacceptable in a scientific paper at all and it can be seen from the provided plots that the results have to be wrong due to untreated boundary conditions, errors introduced by the loading device and estimation of the quantities from the cross-head movement. Moreover, the delta_y is not necessarily a yield point in all cases.

You formulate your conclusions, particularly in case of the energy absorption capabilities, on evaluation from the stress-strain diagram up to 0.4 of overall compressive strain. In this case, why the plots of uniform lattices show the end of the experiment before 0.06 of compressive strain? Conversely, why did you continue the tests with FGM sampels up to 0.4 of compressive strain even if some samples exhibited regions with approx. zero strain, which not only strongly resembles the demonstrated behaviour of the uniform lattices, but also influences the ultimate conclusion of the paper?

No discussion is provided regarding the experimental procedures and data post-processing nor comparison with results assessed on samples with different unit-cell geometries.

Reviewer 2 Report

The introduction section is well prepared. The literature review is sufficient. However in my opinion there is necessary to add the discussion about the mechanical properties of the bulk material itself. In the introduction section, we have some general statements about for example "good mechanical properties", colour, "biocompatibility", but we don't have any value of any parameter. Adding this information will be valuable for comparing the material itself with prepared products (cubic, honeycomb, bamboo).

Additionally, there is should be an extended discussion about manufacturing defects. Of course, we have statements such "Stringing was observed in Fig. 6(e), and it was due to the residual molten plastic from the extrusion head that continued to flow as it moved after the end of one layer’s printing. Rough edge and hole, as shown in Fig. 6(f), were the other main defects that occurred in the manufacturing process" but these type of statements are general. It would be better by adding some scientific discussion about the manufacturing defects to clarify the principle of their obtaining, and proposed methods of their avoiding. This type of discussion should increase the scientific and utilitarian value of the paper.

The text should be checked once again for literal errors and mistakes. In my opinion used English is fine.

Reviewer 3 Report

Congratulations. Mt main concern is about the title. Too long and confusing.

Round 2

Reviewer 1 Report

Although some parts of the manuscript have been improved, its overall scientific merit is still insufficient for publication and I recommend its rejection.

The persisting shortcomings are:

- Inability of authors to comment on RVE of the investigated lattices and discuss its influence on the presented results.
- Instead of providing experimental results regarding the mechanical properties of the material used for 3D printing of the structures at this scale level, the authors have introduced a new section showing some kind of numerical simulation in Abaqus. This section itself is questionable and does not provide the required information (unverified FEA can not be used as a source of fundamental constitutive parameters).
- The authors deal with my finding regarding the reliability of compressive tests, where the strains are derived solely from the cross-head displacement, by providing a reference to the literature. However, such a statement does not clarify scientific soundness of methods considered in the reviewed manuscript.
- Based on my experience with the use of large capacity loading frames for investigation of similar samples, I asked the authors to provide a representative force-displacement curve to show the magnitude of forces to compare with the load-cell characteristics of the testing device, which the authors failed to disclose.
- The authors failed to clarify the reason for selecting the energy absorption parameter as a metric in case of materials, where the overall compressive response consists in cyclic drops of stress to zero level due to brittle failure of structural elements.
- The authors insist on evaluating the elastic modulus by engineering approach from uncorrected cross-head displacement and a region that is apparently non-linear while providing a citation instead of reevaluating the data. This is unacceptable; furthermore, the authors should be aware of the fact that it is common that the elastic properties of cellular solids is established from unloading part of the mechanical response.
- I do not accept the response to point 9, because the other samples show similar behavior with crushing response typical for brittle failure of 3D printed lattices. Thus, the post-processing of data in current state of manuscript shows signs of purposeful interpretation.
- The paragraph provided as a response to point 10 is rather a summary of findings than a proper discussion of results.
